# Targeted Nanoparticle Photodynamic Diagnosis and Therapy of Colorectal Cancer

**DOI:** 10.3390/ijms22189779

**Published:** 2021-09-10

**Authors:** Nokuphila Winifred Nompumelelo Simelane, Cherie Ann Kruger, Heidi Abrahamse

**Affiliations:** Laser Research Centre, Faculty of Health Sciences, University of Johannesburg, P.O. Box 17011, Doornfontein 2028, South Africa; lelopearl@gmail.com (N.W.N.S.); habrahamse@uj.ac.za (H.A.)

**Keywords:** colorectal cancer, photodynamic diagnosis, photodynamic therapy, zinc phthalocyanine, nanoparticle, tumor targeting

## Abstract

Colorectal cancer (CRC) is an aggressive cancer that remains a challenge to diagnose and treat. Photodynamic diagnosis (PDD) and therapy (PDT) are novel alternative techniques, which can enhance early diagnosis, as well as elicit tumor cell death. This is accomplished through photosensitizer (PS) mediated fluorescence and cytotoxic reactive oxygen species activation upon laser light irradiation excitation at specific low and high range wavelengths, respectively. However, the lack of PS target tumor tissue specificity often hampers these techniques. This study successfully fabricated a bioactive nanoconjugate, ZnPcS_4_-AuNP-S-PEG5000-NH_2_-Anti-GCC mAb (BNC), based upon a polyethylene glycol-gold nanoparticle, which was multi-functionalized with a fluorescent PDT metalated zinc phthalocyanine PS, and specific anti-GCC targeting antibodies, to overcome CRC PDD and PDT challenges. The BNC was found to be stable and showed selectively improved subcellular accumulation within targeted CRC for improved PDD and PDT outcomes in comparison to healthy in vitro cultured cells. Additionally, the BNC reported significantly higher late apoptotic PDT-induced CRC cell death rates (34% ***) when compared to PDT PS administration alone (15% *). These results indicated that the improved PDD and PDT outcomes were due to the specific PS accumulation in CRC cells through nanoparticle carriage and bioactive anti-GCC targeting.

## 1. Introduction

Colorectal cancer (CRC) was identified as the fourth deadliest cause of cancer-associated fatalities at the global level [1]. Moreover, CRC is the third most commonly diagnosed cancer worldwide, with a rise in incidence rates each year [1]. Despite the tremendous advancements in currently used diagnostic and treatment options, the overall impact on survival rate is limited, especially when CRC is diagnosed at late advanced stages [1]. Colonoscopy and flexible sigmoidoscopies, some of the definitive diagnostic methods currently utilized, can yield a five year survival rate of around 90% within early localized stages of CRC [2]. However, these conventional diagnostic methods are invasive, nonspecific, and often too inaccurate to diagnose early stage CRC [2]. Moreover, some of the traditional first-line treatment strategies for CRC, such as surgery, radiotherapy, and chemotherapy, are invasive and induce severe unwanted side effects [2]. Additionally, in some cases, CRC is found to be resistant to radio- and chemotherapy once it has metastasized [2]. Therefore, there is a strong need to investigate novel, highly-selective therapeutic modalities that are less invasive, can promote early diagnosis, and enhance treatment efficacy, while causing minimal systemic toxicity and side effects in healthy tissues as compared to conventional treatments [3].

Photodynamic diagnosis (PDD) and photodynamic therapy (PDT) are light-based modalities that can be utilized for the early diagnosis and treatment of CRC [4]. These modalities make use of a photosensitive compound, known as a photosensitizer (PS) [4]. PSs can passively localize in cancer cells and, upon the absorption of visible light at a specific wavelength, they can either exhibit fluorescence (for PDD) and/or convert surrounding cellular oxygen into reactive cytotoxic species, which, in turn, induce cell death [4]. PDD is a non-damaging diagnostic modality that utilizes fluorescence, produced by blue light (340–540 nm) [3]. When a PS has accumulated in malignant cells (and not in normal cells) and is excited with blue light, it can fluoresce, which allows one to easily identify and distinguish tumor cells from healthy cells [3]. PDT, on the other hand, is an alternative therapeutic modality that destroys cancerous tumor cells [3]. Within PDT treatments, after a PS has successfully accumulated in malignant cells, it is exposed to visible red light (620–690 nm) [3]. This red light excites the PS, which in turn reacts with surrounding cellular oxygen producing various cytotoxic reactive radical species, resulting in localized tumor cell destruction [3].

Despite this great potential, the application of PDD-PDT in cancer treatment has some constraints. These restraints are associated with traditional PSs limitations, such as the undesirable accumulation of conventional PSs in healthy cells, lack of target tumor cell specificity, and low absorptions within the red UV light wavelength spectra [5]. Structural modifications, which involve the addition of metal ions in conventional phthalocyanine (Pc) PSs, has substantially improved their photo properties, allowing them to be well suited for the PDD and PDT treatment of CRC [6]. Metalated phthalocyanine (MPc) PSs exhibit improved phototherapeutic effects, higher chemical stability, and photochemical reactivity, as well as excitation at wavelengths within both the blue and red region [6]. Moreover, within PDT applications, the MPc PSs wavelength of excitation ranges above 630 nm, allowing for better tissue light penetration, and improved PDT outcomes are often observed [6]. Studies by Abrahamse and Houreld (2019) reported that metalated zinc phthalocyanine sulfonated mix (ZnPcS_mix_) PS localized within the intracellular lysosomal and mitochondria of in vitro treated CRC cell lines (DLD-1 and CaCo-2) and, thus, favored apoptotic cell death induction under 680 nm of laser irradiation [7]. However, it was previously reported that ZnPcS_mix_ PSs have poor water solubility, which in turn encourages the formation of aggregates and hampers its PDT treatment outcomes [8]. However, chemical modifications, such as the introduction of peripheral (β-position) and non-peripheral (α-position) tetra thiol sulpho substituents within the classical ZnPcS PS structure, can create a far more soluble ZnPc PS derivative, known as zinc sulfothiolphthalocyanine (ZnPcS_4_) (Figure 1) [9].

Within PDT applications, ZnPcS_4_ PSs were reported to be efficient reactive oxygen species (ROS) producers [10]. This is due to their characteristic intense Q-band of peak absorption, observed within the 583, 634, and 674 nm visible infra-red region [11]. Furthermore, the 674 nm wavelength of activation for ZnPcS_4_ PSs can achieve deep-seated tissue penetration and offers more effective PDT treatment outcomes [11,12]. Furthermore, ZnPcS_4_ PSs also possesses a Soret band peak around 330 to 350 nm (visible blue light) of excitation [10]. This makes them ideal PSs to utilize within PDD applications, since they can emit fluorescence for photodetection [11,12]. Additionally, due to the enhanced permeation and retention (EPR) effect, ZnPcS_4_ PSs are able to passively localize in cancerous cells and, thus, can be contained more effectively [3]. All these beneficial factors are suggestive that ZnPcS_4_ PSs can possibly offer the capabilities of simultaneous PDD and PDT treatment of CRC.

However, even with ZnPcS_4_ PS tetra thiol sulpho substituent modifications, in order to achieve significant PDD and PDT treatment outcomes, the solubility and active tumor cellular location of this PS in CRC needs to be achieved [11].

Thus, researchers have focused on several biomarker-conjugated PS nanocarriers in order to overcome the shortfalls that conventional PS face [13]. From the various types of nanocarriers, which were investigated within photo-diagnostic and therapeutic applications, gold nanoparticles (AuNPs) were found to be the most efficient [14]. AuNPs offer several advantages that have rendered them suitable for PDD and PDT applications [14]. AuNPs have a large surface, which allows for modification, they are size tuneable, and they are biocompatible within in vivo environments [14]. Furthermore, AuNPs have the ability to serve as bioconjugate drug carriers when bound with specific targeting ligands or biomarkers [14,15]. This, in turn, can promote active uptake and cellular localization of the drugs they carry for maximum treatment outcomes [14,15]. Various studies have noted that the utilization of biomarker-conjugated PS nanocarriers improves the specific and actively targeted uptake of PSs in tumor cells, which in turn promotes PDD and PDT treatment outcomes [8]. Thus, active bio-targeting, whereby AuNPs surfaces are conjugated with antibody biomarker targeting moieties for specifically targeted cancerous cells recognition and improved subcellular localization of drugs, has attracted great attention [15].

Studies by Danaee et al. (2017) reported that the transmembrane receptor protein, guanylyl cyclase C (GCC), was selectively and specifically overexpressed in CRC cells and thus can potentially be utilized as a specific biomarker to mediate anti-antibody targeted CRC PS uptake and retention within photo-diagnostic and treatment therapies [15,16].

Therefore, this study fabricated ZnPcS_4_ PS functionalized AuNPs, which were conjugated to anti-GCC monoclonal antibodies (mAbs), in order to form a final multifunctional tumor-targeted bioactive nanoconjugate (BNC) (Figure 2). This study also went on to investigate if this BNC could actively and specifically enhance the ZnPcS_4_ PS uptake within GCC-positive in vitro cultured CRC cells and attempt to achieve effective and efficient PDD and PDT treatment outcomes (Figure 2).

## 2. Results

### 2.1. ZnPcS_4_ PS and PDT Dose Response Assays

The optimal concentration dose (ICD_50_) for ZnPcS_4_ PS PDT dose response were performed in order to determine the required concentration of ZnPcS_4_ PS that was needed within the final BNC for effective PDD or PDT outcomes. The favorable ICD_50_ of the ZnPcS_4_ PS post-PDT was found to be 0.125 µM. The trypan blue exclusion cell viability assay and lactate dehydrogenase (LDH) cytotoxicity and membrane integrity assays results can be found in the Appendix A.

### 2.2. Molecular Characterization of the Final Multifunctional Tumor-Targeted Bioactive Nanoconjugate (BNC)

#### 2.2.1. UV Visible Spectroscopy

The UV visible spectra of the final BNC noted a lowered presence of two major characteristic Q-bands at 634 and 674 nm, as well as a lowered characteristic Soret absorption peak at 340 nm, confirming the presence of the ZnPcS_4_ PS within the final BNC (Figure 3) [11,17]. Additionally, this absorption spectrum confirmed the incorporation of the PEGylated AuNPs within the final BNC, as a characteristic lowered absorption band at 520 nm was observed [18]. The major absorbance bands at 520 nm, 634 nm, and 673 nm of the BNC, compared to the ZnPcS_4_ PS and AuNP-S-PEG5000-NH_2_ control peaks, did display a slight broadening and decrease in presence, suggesting that quenching had possibly occurred during binding [8,19]. The quenching of PSs when binding them to NPs to improve PS uptake is usually due to self-aggregation [8]. Even though self-aggregation tends to reduce singlet oxygen generation within PDT applications, it can improve the generation of other radical species, such as hydroxyl and peroxyl radicals, which also promote tumor cell death [19]. Furthermore, since the NP was also metalated, it would also promote PDT-induced hypothermia reactions and, as a result, the final bound BNC would still be capable of producing favorable PDT treatment outcomes [19].

When the major absorption peak fold fall of the final BNC were compared to that of the ZnPcS_4_ PS and AuNP-S-PEG5000-NH_2_ control bands, and it was determined that the BNC consisted of 0.95 × 10^15^ particles/mL AuNPs bound to 35 µM of ZnPcS_4_ PS (Figure 4) [20]. Overall, these results demonstrated the successful conjugation between ZnPcS_4_ PS and AuNP-S-PEG5000-NH_2_ and, since the PSs absorption and emission peaks remained intact, these results suggested that it retained its photochemical properties.

To ensure successful covalent binding of the anti-GCC mAb within the final BNC UV visible protein, direct absorption spectra were assessed (Figure 4). The final BNC displayed amine functional groups at the 195 nm absorption peak. These findings suggest that the n’ terminus anti-GCC mAb sites within the BNC were well-retained for active targeting, and that only their c’ terminus sites had covalently bound with the AuNP-S-PEG5000-NH_2_ [21]. Thus, the bound anti-GCC mAbs within the final BNC were well orientated for actively specific antigen-antibody recognition in order to enhance PS delivery ZnPcS_4_ in CRC cells [21]. Additionally, the UV vis absorption spectra of the final BNC had higher peaks within the 250 to 280 nm range in comparison to the AuNP-S-PEG5000-NH_2_ and ZnPcS_4_ PS control peaks, suggesting the presence of disulphide bond formation [17]. The presence of this disulphide bond probably occurred during the ligand exchange process, since it was performed in the presence of air overnight [17,22]. This caused the ZnPcS_4_ PS to form a dimer (ZnPcS_4_)_2_ and made the final BNC assembly: AuNP-S-PEG5000-NH_2_-Anti-GCC mAb+ ZnPcS_4_ + (ZnPcS_4_)_2_ [16,21]. Moreover, within the protein direct UV 280 nm spectral region, the final BNC noted a 6.47 absorbance fold fall when compared to that of the free anti-GCC mAb, and, thus, confirmed that 30.91 µg/mL of anti-GCC mAb was successfully bound within the final BNC (Figure 4) (Appendix A) [23].

Since the above molecular characterization results noted that the final BNC consisted of 30.91 µg/mL of anti-GCC mAb, which were successfully bound to 0.95 × 10^15^ AuNP-S-PEG5000-NH_2_ particles/mL, containing 35 µM of ZnPcS_4_ in PBS, it meant dilution was required because the PS PDT dose response assays indicated that the favorable ICD_50_ of ZnPcS_4_ PS post-PDT for CRC treatment was 0.125 µM. Thus, the final BNC required a 280-fold dilution correction factor to dilute the 35 µM ZnPcS_4_ PS concentration to the recommended ICD_50_ concentration of 0.125 µM for favorable further experimentation. As such, within the final BNC PDD and PDT response assays, the cell culture plates received 0.11 µg/mL of anti-GCC Ab, which was successfully bound to 3.39 × 10^12^ AuNP-S-PEG5000-NH_2_ particles/mL conjugated to a 0.125 µM of ZnPcS_4_ in 1.0 mM PBS (*w*/*v*).

#### 2.2.2. FT-IR Spectroscopy

Fourier transform infrared spectroscopy (FT-IR) analysis was performed to confirm the successful ligand exchange (Au-S) and disulphide dimer formation of the ZnPcS_4_ PS within the final BNC. Additionally, FT-IR was used to identify the formation of amide bonds between the amine (NH) functionalized group on the AuNPs and the activated c’ terminus of the anti-GCC mAb. Confirmatory results, with control comparisons, are presented in Appendix A.

#### 2.2.3. Dynamic Light Scattering and Zeta Potential

The measurement of the zeta potential (ζ) and dynamic light scattering (DLS) was performed to determine the final BNC’s surface charge and size. The results are available in Appendix A.

ZP results reported the final BNC had a mean Z-average diameter of 57.18 ± 3.04 nm. These results suggest that it was small enough to be considered an active nano drug-carrying system, with a good possibility of cellular uptake and absorption [24]. The ZP value of the final BNC was 36.5 ± 2.6 mV, and it was highly stable with a slightly positive charge [25]. This ZP value suggested that the final BNC should remain constant and stable within an in vivo environment, as well as be retained more selectively within negatively charged tumor cells [25,26].

DLS peak analysis found the final BNC to be homogenously pure, spherical in shape, with no aggregation [26]. The DLS PDI value of the final BNC was found to be 0.353, signifying it was monodispersed and consisted mostly of singular sized particles [26].

Overall, these results suggested that all three constituents were successfully bound together to form one single final BNC molecule [27,28].

### 2.3. Subcellular Localization Assays

To investigate the subcellular localization and active targeting affinity of the final BNC within anti-GCC mAb positive CRC CaCo-2 cells, versus normal human cells (WS1 fibroblasts), images were captured with immunofluorescent microscopy. Green immunofluorescent ICAM-1/FITC membrane staining, blue immunofluorescent DAPI nuclei staining, and ZnPcS_4_ PS red 673 nm fluorescence signals from various experimental and control group images were compared. The cells only control for both CRC CaCo-2 and WS1 fibroblast cells reported no background red autofluorescence. Thus, it was safe to assume that for the remaining control and experimental groups that the red fluoresce was only due to the presentable detection of the ZnPcS_4_ PS (Figure 5 and Figure 6).

With reference to Figure 5, CRC cells were noted to have a slight passive subcellular uptake of the ZnPcS_4_ PS when it was administered alone. These results coincided with the findings reported by Roguin et al. (2019), highlighting that ZnPc is a stable compound, however, it exhibits only slightly selective passive accumulation in tumor tissues [29]. This limited passive accumulation in tumors was found to be due to aggregation, which affects the PSs accessibility and uptake in neoplastic tissues [29]. When comparing this image to CRC cells that received the same concentration of ZnPcS_4_ PS, however, conjugated AuNPs showed an improved PS passive uptake. Studies by Hong et al. (2016) reported that NPs significantly enhanced the solubility of PSs, as well as limited their aggregation and, thus, increased their passive localization more selectively [13]. However, when comparing both of the above images to CRC cells, which received the same ZnPcS_4_ PS concentration in the final BNC form, an enhanced affinity and active accumulation of the PS was observed. Several lines of evidence have emphasized that, when targeting moieties (which correlate with overexpressed cancer cell receptors) are correctly attached to PS nanocarrier platforms, the PS uptake is enhanced [30]. Studies by Hodgkinson et al. (2017) stated that PS bound NPs, which have conjugated targeting ligands, and showed higher PS uptake and retention, with improved PDT outcomes, due to target tumor cells actively absorbing higher concentrations of PSs via endocytosis [30]. Overall, these results indicate that the final BNC facilitated enhanced ZnPcS_4_ PS retention in CRC cells via the specific PSs anti-GCC mAbs nanoconjugates’ selective and active targeting abilities.

As can be observed in Figure 6, normal human fibroblasts WS1 cells treated with ZnPcS_4_ alone revealed some absorption of the PS. These findings suggest that within conventional CRC PDT treatments, normal human cells would be affected, meaning unwanted side effects could be observed. Whereas normal human fibroblasts WS1 cells, treated with AuNP-S-PEG5000-NH_2_-ZnPcS_4_, showed a lesser absorption of the PS when administered at the same administered concentration. These results suggest that AuNPs promote a more passive uptake within cancer tissues and, thus, CRC PDT treatment outcomes would have lessened side effects on normal surrounding tissues when PSs are administered in combination with AuNPs [13]. However, experimental normal human fibroblast WS1 cells treated with the final BNC showed no detectable subcellular localization of the PS. These findings indicated that the final BNC had no actively specific affinity for healthy cells (such as fibroblasts). These results suggest that, if the final BNC was to be applied within CRC PDT treatments, the treatment would remain localized to CRC only and no unwanted side effects would be found within healthy surrounding tissues. Lastly, these findings suggested that, due to the specific subcellular localization and active targeting affinity the final BNC had within CRC cells only, and not within normal human cells, it could possibly be accurately utilized within PDD assays to label and identify CRC.

### 2.4. Photodiagnostic Assays

To investigate the photodiagnostic abilities of the final BNC within anti-GCC mAb positive CRC CaCo-2 cells, versus negative anti-GCC mAb normal human cells (WS1 fibroblasts), images were captured using immunofluorescent microscopy. Green immunofluorescent ICAM-1/FITC membrane staining and ZnPcS_4_ PS blue 340 nm PDD fluorescence signals, from various experimental and control group images, were compared to one another. The cells only control for both CRC CaCo-2 and WS1 fibroblast cells reported no background blue autofluorescence and, thus, it was assumed that, for the remining control and experimental groups, the blue fluoresce was only due to the presence of the ZnPcS_4_ PS (Figure 7 and Figure 8).

As shown in Figure 7, the blue fluorescence intensity of the final BNC was markedly increased relative to all the other CRC control groups. Additionally, the free ZnPcS_4_ PS only displayed a lesser degree of selective passive blue fluorescence intensity accumulation in tumor tissues then when compared to the CRC control groups that received AuNP-S-PEG5000-NH_2_-ZnPcS_4_. These findings suggest that the AuNPs significantly improved the solubility of PSs and increased its selective passive localization into the tumor cells at a far more superior rate than free ZnPcS_4_ PS administration alone [13]. However, since the experimental group of CRC cells, which received the final BNC, revealed the highest accumulation of ZnPcS_4_ PS, it can be stated that the conjugated anti-GCC mAbs must have been responsible. Previous studies by Kruger and Abrahamse (2019), as well as Danaee et al. (2017), also noted the successful active uptake and heightened retention of PSs in CRC cells when targeting was specifically directed towards GCC overexpressed cellular surface receptors [15,16].

Lastly, regarding Figure 8, WS1 normal human cells control groups, which received free ZnPcS_4_ PS only, reported a concentrated PS uptake. CRC control groups, which received AuNP-S-PEG5000-NH_2_-ZnPcS_4_, noted a slight uptake of the PS, whereas the experimental group, which received the final BNC, showed no selective or active PS accumulation. These findings suggest that the final BNC had no actively specific targeting affinity for normal human cells and could be successfully utilized in PDD applications to distinguish between normal and in vitro cultured CaCo-2 CRC cells. Studies by Waldman et al. (1998) support these findings, wherein they stated that GCC overexpression was specific to CaCo-2 cell cultures only, and that, since it was not found in healthy extra-intestinal tissues (such as WS1 fibroblasts), it could be utilized as a highly specific diagnostic and identification biomarker for CRC [31].

### 2.5. Morphological Assessment Post-PDT

The cellular morphological effects in response to the final BNC PDT treatment, within the different control and experimental groups, were examined and digitally captured via inverted light microscopy (Figure 9).

The control group of CRC cells only maintained their morphological distinctive characteristics of epithelial-like bodies with tight junctions [32,33]. Control cells treated with laser irradiation only showed no significant changes in their cellular morphology. This observation was indicative that laser irradiation alone was not phototoxic to CRC cells. Studies conducted by Sekhejane et al. (2014) also noted that laser irradiation at 673 nm, with a fluence of 10 J/cm^2^, induced no phototoxic side effects within in vitro cultured CaCo-2 [34]. Within control groups of CRC cells, which received ZnPcS_4_ PS alone, AuNP-S-PEG5000-NH_2_-ZnPcS_4_, or the final BNC without laser irradiation, the cells remained intact with distinctive viability and morphological features. These findings suggest that neither the PS alone, PS + AuNPs, or the final BNC induced any form of dark cytotoxicity and, thus, the PS remained inactive in the absence of light [34]. In contrast, PDT experimental groups, which received ZnPcS_4_ PS alone, AuNP-S-PEG5000-NH_2_-ZnPcS_4__,_ or the final BNC with laser irradiation, showed notable changes of morphological structural damage, such as a loss of cellular shape, detachment, and free-floating cells.

Nevertheless, the most notable morphological cellular destruction was observed in PDT experimental groups, which received the final BNC, in comparison to the other PDT groupings. Minimal cellular damage was observed in PDT experimental groups, which received ZnPcS_4_ PS alone. However, a slightly more damaging effect was observed in the PDT experimental group that received AuNP-SPEG5000-NH_2_-ZnPcS_4_. Hong et al. (2016) reported that when PSs were conjugated to AuNPs, their passive uptake in tumor cells was improved upon via the EPR effect and, when activated with laser irradiation, the PDT effects were more cytotoxic than that of free PS PDT treatment alone [13]. The study’s reasoning for this improved PS-AuNPs conjugates enhanced PDT phototoxicity was attributed to the enhanced passive PS localization in tumor cells, as well as the metalated photothermal destruction the AuNPs induced [35]. Taking these attributing factors into account, with the enhanced specific active targeting abilities of the final BNC, it would explain why the most cellular damage was observed in PDT experimental groups that received the final BNC. These findings suggest that the final BNC was able to more specifically and actively enhance the PS uptake in CRC cells due to its direct targeting abilities and, as a result, the PDT treatment outcomes were significantly enhanced [15].

### 2.6. Cellular Death Pathway Detection Assay Post-PDT

An Annexin V-FITC/PI cell death staining kit was used for the flow cytometry detection and quantitation of CRC control and experimental cells post-PDT, which either remained viable, underwent early/late apoptosis, or suffered necrosis. Control CRC cells depicting the various modes of cell death were used to gate and statistically compare the final result outcomes (Figure 10).

Within Figure 10, it can be observed that control groups of CRC cells treated with laser irradiation alone presented no significant cell death changes. These observations suggest that laser irradiation alone had no phototoxic side effect on CRC cells [34]. Non-irradiated control groups of CRC cells, which received ZnPcS_4_ PS, AuNP-S-PEG5000-NH_2_-ZnPcS_4_, or the final BNC, only noted slight insignificant increases in apoptotic forms of cell death, with decreases in cellular viability. These findings suggest that the same concentration of ZnPcS_4_ PS, applied throughout experimentation alone, or in conjugation with AuNPs, or in the final BNC form, remained inactive in the absence of laser irradiation and exhibited no dark toxicity. Similar findings were reported by Stuchinskaya et al. (2011) within the targeted PDT treatment of in vitro cultured breast cancer cells using antibody-phthalocyanine AuNP conjugates, whereby the PS, as well as the mAb and AuNPs, exhibited no dark cytotoxicity [17].

Nevertheless, PDT CRC experimental groups, which received ZnPcS_4_ PS alone, AuNP-S-PEG5000-NH_2_-ZnPcS_4__,_ or the final BNC with laser irradiation, revealed significant forms of apoptotic cell death, with decreases in cellular viability. However, PDT irradiated CRC experimental groups, which received ZnPcS_4_ PS alone, did not exhibit as much significant late apoptotic cell death (15% *) as when compared to the same groups, which received AuNP-S-PEG5000-NH_2_-ZnPcS_4_ (21% **) (Figure 10). Furthermore, when comparing these two groups to the CRC experimental group, which received the final BNC plus laser irradiation, the most significant forms of late apoptotic cell death (34% ***) was found, with a significant increase of necrotic cell death (16%*) (Figure 10).

Accruing evidence showed that cells undergoing early apoptosis are often autophagic and have the potential to rejuvenate. Thus, early phases of apoptotic cell death are not ideal in PDT cancer treatments, since cells have the potential to recover, which can leading to tumor re-occurrence [36]. As a result, the most favorable form of cell death, in relation to successful PDT treatment outcomes, is either late apoptotic or necrotic cell death modes, whereby cancer cells are destroyed beyond recovery [37,38].

Since second-generation PSs (such as ZnPcS_4_), when administered alone, are hydrophobic and tend to aggregate under physiological conditions, this drastically limits their uptake in tumor cells, which in turn hinders PDT treatment outcomes [35]. A study by Kruger and Abrahamse (2018) could explain why the ZnPcS_4_ PS alone PDT treatment did not exhibit as much significant late apoptotic cell death (15% *) as when compared to the same experimental groups receiving AuNP-S-PEG5000-NH_2_-ZnPcS_4_ (21% **) (Figure 10) [35]. Since studies by Hong et al. (2016) showed that, when PSs were combined with AuNPs, their passive uptake and solubility in tumor cells was improved upon via the EPR effect and when activated with laser irradiation, more favorable PDT-induced late apoptotic forms of cell death were observed [13]. Furthermore, studies by Kruger and Abrahamse (2018) have noted that AuNPs tend to enhance overall PDT cellular destruction outcomes through contributory metalated photothermal destruction [35]. Thus, the higher late apoptotic cell death of 21%** being noted in the PDT irradiated experimental groups of CRC cells, which received 0.125 µM ZnPcS_4_-AuNP-S-PEG5000-NH_2_, was attributed to the AuNPs passive promotion of ZnPcS_4_ PS uptake and contributing photothermal annihilation (Figure 10).

Nevertheless, the most noteworthy cytotoxic forms of PDT-induced tumor cell death was found within the PDT CRC experimental groups that received the final BNC and irradiation (Figure 10). In comparison to all other control groups, PDT CRC experimental groups that received the final BNC and irradiation noted the most significant decreases in cellular viability (31% ***), with highly significant late forms of apoptotic cell death (34% ***) and increased rates of necrosis (16% *). These auspiciously enhanced forms of PDT-induced cell death by the final BNC were probably due to the fact that it had an actively selective targeting mAb attached to its PS Au-based nanocarrier. These findings are supported by studies conducted by Kruger and Abrahamse (2019), whereby the most significant PDT-induced forms of apoptotic and necrotic cellular destruction was observed when bound AuNP-PSs were conjugated to targeting ligands [15]. The reasoning for these outcomes was attributed to the fact that the cancer directed targeting mAbs promoted PS localization within tumor cells in a far more specifically targeted way than when compared to PS administration alone [15]. This improved actively targeted PS nano-delivery system enhanced PS concentration within tumor cells, and the overall production of PDT-induced cytotoxic species was heightened [15]. Additionally, the AuNPs were metalated and so contributed to PDT-induced tumor targeted photothermal destruction [15]. Thus, the findings from this study suggested that the final targeted BNC was most effective in terms of PS uptake and PDT-induced CRC cancer cell obliteration.

## 3. Discussion

CRC is among one of the most lethal malignancies worldwide, and current conventional diagnostic and therapeutic modalities are often non-specific, invasive, resistant, and induce toxic side effects in healthy tissues [39,40]. According to studies, the early diagnosis of CRC offers the probability of selecting an effective treatment modality, which could increase a patient’s survival rate to approximately 90% [41]. However, due to the typical shortfalls of traditional diagnosis modalities, such as a lack of sensitivity, early diagnosis of CRC is limited and, as such, the prognosis of patients is often relatively poor [41]. Thus, poor tumor selectivity, non-specificity, resistance, and a lack of targeted drug distributions are major challenges that classical CRC diagnostic and treatment strategies tend to suffer from [42,43]. Hence, CRC research has become more focused on alternative diagnostic and treatment modalities that are believed to be more advantageous in relation to specificity and repeatability, as well as cause limited unwanted side effects [30,42].

Along this line, unconventional diagnostic and treatment strategies for cancer, such as PDD and PDT, have attracted considerable attention [42]. The reasoning for this is that PDD and PDT modalities were shown to possess clear advantages over classical modalities, such as minimal invasiveness, high selectivity towards cancerous tissue, and fewer side effects after treatment [44]. According to a recent review performed by Kawczyk-Krupka et al. (2020) and Gunaydin et al. (2021), clinical trials in relation to the utilization of PDT and PDD for diagnosis and the treatment of throat and neck cancers, as well as selected forms of liver, pancreatic, breast, and prostate cancer, were promising [45,46]. However, in terms of successful PDD and PDT combinative clinical trials, with reference to CRC, there have currently been none reported to date [45,47].

Nevertheless, in vitro and in vivo studies into the utilization of PDD and PDT for the diagnosis of CRC remain ongoing. Studies by Kawczyk-Krupka et al. (2018) evaluated the influence of 5-ALA PS mediated PDD and PDT within CRC cells [48]. The study reported that sublethal PDT doses did have the potential to inhibit the processes of proliferation, migration, angiogenesis, and metastasis, however, the 5-ALA PS PDD fluorescence intensity was too low to facilitate differential diagnosis [48]. Lu et al. (2016) encapsulated a small-molecule immunotherapy agent, which inhibits indoleamine 2,3-dioxygenase (IDO), into a chlorin-based nanoscale metal-organic framework and the post-PDT treatment reported successful systematic anti-tumor immunity in CRC models [49]. Unfortunately, this study noted that the chlorin-based PS reported no detectable fluorescence within the blue wavelength region for effective PDD investigations [49]. However, a study performed by Salgado et al. (2016) investigated the in vitro photodynamic application of Hexvix (which is FDA-approved for the PDD diagnosis in bladder cancer) [50]. The study reported the successful photo detection of SW480 CRC cells when blue light was applied, however, only noted a slight red light induced phototoxicity when red light was applied [50]. This study noted that, in order to investigate combinative photodynamic applications for effective CRC PDD and PDT modalities, broad range nano-targeted PSs, which can be activated within low fluorescent ranges and high reactive wavelengths, require investigation [50]. These types of nano-targeted PSs investigations would improve selective PS uptake for sensitive detection, as well as allow for elevated levels of PDT annihilation, since CRC is an aggressive form of cancer [50].

Recent studies by Singh et al. (2018) reported that, due to the remarkable properties AuNPs possess, they have long been considered as a potential PS nanocarrier systems for improved photodynamic diagnostic and treatment outcomes [14]. AuNPs have a high surface-area-to-volume ratio, surface plasmon resonance, and surface chemistry, which can be functionalized with various targeting biomarkers to promote PS uptake [14]. Additionally, they are non-toxic and non-immunogenic in nature and easily promote the passive uptake and retention of PSs via the EPR effect [14]. Moreover, within PDT application, AuNPs promote tumor destruction since they produce photothermal energy when activated with red light [14]. Thus, in relation to the treatment of cancer through tumor detection, drug delivery, imaging, photothermal, and photodynamic therapy, AuNPs are most definitely utilizable nanocarriers within research investigations. Recent studies by Khan et al. (2021) reported the successful application of AuNPs for the phototreatment and diagnosis of CRC, however, they noted that the study’s outcomes could be improved upon if the AuNPs were further modified with CRC bio-targeting markers [51].

Thus, in an effort to further circumvent the non-specificity of PSs and enhance their cellular accumulation, considerable efforts were also devoted towards the synthesis and characterization of bio-conjugate markers, which can increase host PS accumulation specificity, in targeted cancer tumors [13]. Studies by Calavia et al. (2018) reported the improved PDT treatment of in vitro cultured CRC cells when a combination of ZnPcS PS conjugated to AuNPs was administered due to improved PSs passive uptake [52]. Moreover, studies by Sehgal et al. (2013) investigated the conjugation of ZnPcS PS to mAbs, directed against carcinoembryonic antigen (CEA), to improve targeted PS uptake within in vitro cultured CRC cells [53]. Their results showed that the cellular CRC uptake of the bioconjugate was 37 times higher than compared to free ZnPcS administration [53]. Furthermore, this study noted that ZnPcS-anti-CEA PS conjugate produced an intense fluorescence within the blue light region, with negligible phototoxicity [53]. Overall, this study proved that the ZnPcS-anti-CEA PS conjugate was highly selective in targeting CEA overexpression in CRC [53]. The findings from this study suggest that the ZnPcS-anti-CEA PS conjugate could be particularly useful for PDT treatment, as well as be utilized as a PDD agent for the fluorescent surveillance of CRC cells, however, investigations remain ongoing [53].

Investigations performed by Waldman et al. (1998), examined the heterogeneity of GCC expression within in vitro cultured CaCo-2 cell lines and noted a prominent overexpression of GCC [31]. Furthermore, this study reported that extra-intestinal tissues in healthy adult humans showed no expression of GCC [31]. The study concluded that GCC could possibly be utilized as a highly specific identification biomarker for the successful detection of CRC in diagnostic applications, as well as serve as a precise targeting ligand for enhanced drug delivery in CRC treatments [31].

In this study, we aimed to combine an efficient cytotoxic oxygen-generating ZnPcS_4_ PS with CRC-specific targeting mAbs anti-GCC on the surface of heterobifunctional PEG amine stabilized AuNPs. The objective of this study was to try and enhance the active ZnPcS_4_ PS uptake specifically in CRC cells, so that the PDD and PDT treatment of CRC tumors could be improved upon. Since this type of GCC overexpression with in vitro cultured CRC cells had never been investigated before, wherein an AuNP ZnPcS_4_ PS-targeted carrier was utilized for the combinative PDD and PDT treatment of CRC, the novelty and need for this study was attractive.

In summary, a final BNC, which consisted of conjugated ZnPcS_4_ PS-PEGlated AuNPs with specific CRC-targeting anti-GCC mAbs, was successfully developed. UV visible and FT-IR spectroscopy molecular characterization results confirmed successful Au-S ligand bond exchange between the ZnPcS_4_ PS-PEGlated AuNPs, as well as noted effective covalent amide bonding between the c’ terminus of the anti-GCC mAbs and PEGlated AuNPs. Furthermore, these assays reported effective anti-GCC mAbs orientation on the PEGlated AuNPs, noting that the amine n’ terminus was still active for specific CRC-targeting. Additionally, the final BNC UV visible spectroscopy results displayed characteristic lowered ZnPcS_4_ PS signature peaks at 340 nm and 674 nm wavelength regions, suggesting that even though some quenching and aggregation occurred, it should still be able to effectively function within PDD and PDT assays [8,19]. Lastly, the final BNC UV visible spectroscopy results revealed the presence of the signature 520 nm PEGlated AuNPs band, suggesting that the NPs photothermal function remained intact and was able to contribute to PDT-induced annihilation of CRCs [51]. DLS and ZP characterization results of the final BNC reported high stability and monomodal particle size distribution patterns, suggesting it was stable and small enough to be accepted as an active nano-drug carrying platform [24,25,26,27].

With reference to the subcellular localization and PDD assays, the final BNC noted significantly enhanced ZnPcS_4_ PS uptake in CRC cells when compared to the control group of ZnPcS_4_ PS administered alone. While the AuNP-S-PEG5000-NH_2_-ZnPcS_4_ CRC control group did note a higher passive accumulation of the PS, when compared to ZnPcS_4_ PS administration alone, the most significant and specifically localized uptake was observed in CRC cells that received the final BNC. Furthermore, within WS1 subcellular localization assays the final BNC reported no ZnPcS_4_ PS specificity and selective uptake in cells. These findings are probably owing to the attachment of anti-GCC mAbs, since the final BNC specifically bound with targeted CRC cells only and not with normal WS1 cells. These findings are in agreement with studies conducted by Gao et al. (2017), which reported appreciable anti-proliferative effects within in vivo tumor models, with negligible effects on normal tissues when PSs were encapsulated in an active targeting moiety [54]. Overall, these findings confirmed that the final BNC produced a highly selective and accumulative ZnPcS_4_ PS-targeting affinity in CRC cells only, as well as produced a distinguishable PPD blue-fluorescent signal that could successfully distinguish between normal and in vitro cultured CaCo-2 CRC cells.

In relation to PDT CRC assays, the control groups that received laser irradiation alone noted no phototoxic side effects and non-irradiated control groups, which received ZnPcS_4_ PS, AuNP-S-PEG5000-NH_2_-ZnPcS_4_, or the final BNC, reported no significant forms of cytotoxic cell death. These findings suggested that the final BNC, in its conjugated form or when in it was within its singular component form, exhibited no phototoxicity or dark toxicity. All PDT CRC experimental groups that received laser irradiation did report some form of significant late apoptotic cell death and decreased cellular viability. However, the PDT CRC experimental groups, which received the final BNC plus laser irradiation, reported the most significant forms of late apoptotic cell death (34% ***) when compared to ZnPcS_4_ PS alone (15% *) and AuNP-S-PEG5000-NH_2_-ZnPcS_4_ (21% **) groups. The improved and more significant late apoptotic PDT CRC results for the experimental group that received AuNP-S-PEG5000-NH_2_-ZnPcS_4_ of 21% **, in comparison to the same grouping, which received ZnPcS_4_ PS alone of 15% *, was attributed to the presence of the AuNPs. Studies by Kruger and Abrahamse (2018) support this finding, stating that PSs, when conjugated to AuNPs, enhanced the overall cytotoxicity outcomes in PDT treatments as compared to PS administration alone, since they promoted a more passive uptake of PSs and contributed to photothermal destruction [35]. However, since the PDT CRC experimental groups, which received the final BNC plus laser irradiation, reported the most significant forms of late apoptotic cell death (34% ***) in comparison to all other groupings, these findings suggest that the conjugated anti-GCC mAb, actively enhanced PS uptake and retention in combination with AuNPs passivation, thus producing the most favorable form of late apoptotic cell death. Similar findings were reported by Stuchinskaya et al. (2011), whereby they investigated the targeted PDT treatment within in vitro cultured breast cancer cells [17]. The study reported that the combinative final antibody-phthalocyanine PS AuNP conjugate noted the most improved PS active uptake and retention in cancer cells, when compared to its singular components [17]. Furthermore, the study went on to report that within PDT assays the final antibody-phthalocyanine PS AuNP conjugate reported the most significant forms of apoptotic cell death in cultured cells [17]. The researchers attributed these findings to the PSs mAbs selective breast cancer nano-targeting abilities, which significantly enhanced the PS absorption in tumors and drastically improved overall PDT treatment outcomes [17]. Thus, the improved PDT CRC treatment outcomes produced by the final BNC was attributed to the fact that it actively and specifically enhanced the uptake of ZnPcS_4_ PS through direct anti-GCC mAb targeting and, with AuNP photothermal induction, was able to trigger the most significant forms of late apoptotic cell death in CRC [15].

Overall, this study demonstrated that the final BNC efficiently promoted the diagnostic and therapeutic performance of PDD and PDT within in vitro CRC.

## 4. Materials and Methods

### 4.1. Cell Culture

Human colon cancer cells (CaCo-2 Cellonex Cat SS1402 CCAC-FL; CCAC-C) and WS1 human skin fibroblasts were obtained from the American Type Culture Collection (ATTC CRL-1502). The utilization of Caco-2 cells as the GCC positive and WS1 cells as GCC negative control was supported according to studies performed by Waldman et al. (1998) [31]. GCC is a transmembrane surface receptor restricted to intestinal epithelial cells from the duodenum to the rectum/colon and appears to specifically be overexpressed in colorectal adenocarcinomas [31]. The study went on to examine the heterogeneity of GCC expression in eight human colorectal carcinoma cell lines in vitro using RT-PCR analysis and noted a prominent overexpression of GCC within CaCo-2 cell cultures [31]. Furthermore, this same study noted that extra-intestinal tissues in healthy adult humans (such as WS1 fibroblasts) showed no expression of GCC [31]. Thus, it was concluded that GCC could be utilized as a highly specific diagnostic and identification biomarker for the staging and postoperative surveillance of patients with CRC [31].

Caco-2 and WS1 cells were cultured in DMEM media, enriched with 10% fetal bovine serum (FBS), 4 mM sodium pyruvate, 4 mM L-glutamine, 2.5 g/mL amphotericin B, and 100 U Penicillin 100 g/mL streptomycin solution, and maintained at 37 °C in 5% CO_2_ and 85% humidified air. Once confluent, monolayers of the cells were harvested using TrypLE Select™ and resultant cellular pellets were re-suspended in fresh culture media. Thereafter, cells were seeded at a density of 6 × 10^5^ cells/3 mL of supplemented media in 3.4 cm diameter culture plates and incubated for 4 h to allow for cellular attachment before conducting experiments.

### 4.2. ZnPcS_4_ PS and PDT Dose Response Assays

A stock solution of 0.5 mM ZnPcS_4_ PS (Santa Cruz^®^ Biotechnology sc-264509A) was diluted in 4 mL of 1.0 mM phosphate buffered saline (PBS) to make a working concentration of 125 µM. This working solution was further diluted with 1.0 mM PBS, to make concentrations of 0.0312, 0.0625, 0.125, and 0.25 µM ZnPcS_4_ PS. These varying concentrations of ZnPcS_4_ PS were administered to in vitro cultured CRC CaCo-2 cells in order to determine the lowest inhibitory concentration (ICD_50_) that was required to induce 50% cytotoxicity 24 h post-PDT irradiation. The ZnPcS_4_ PS ICD_50_ was determined post-PDT using the trypan blue exclusion and the lactate dehydrogenase cellular cytotoxicity and membrane integrity biochemical assay (Appendix A).

The ICD_50_ ZnPcS_4_ PS PDT dose response assay was necessary in order to determine the required concentration of ZnPcS_4_ PS that was needed to be conjugated within the final BNC for effective PDD or PDT outcomes. The favorable ICD_50_ of the ZnPcS_4_ PS post-PDT was found to be 0.125 µM (Appendix A).

### 4.3. Cellular Treatment and PDD/PDT Laser Irradiation

At 4 h post incubation, cell culture plates were divided into several control and experimental groups for ZnPcS_4_ PS PDT dose response assays and BNC PDD or PDT assays. The control and experimental groups within ZnPcS_4_ PS PDT dose response assays had their old cell culture media removed and replaced with 3 mL concentration-adjusted cell culture media containing the varying concentrations of ZnPcS_4_ PS. The control and experimental groups within BNC PDD or PDT assays had their old cell culture media removed and replaced with 3 mL concentration adjusted cell culture media containing varying components: 0.125 µM ZnPcS_4_, 0.125 µM ZnPcS_4_ + 3.39 × 10^12^ AuNP-S-PEG5000-NH_2_ particles/mL, 0.125 µM ZnPcS_4_ + 0.11 µg/mL anti-GCC Ab or BNC (0.11 µg/mL anti-GCC Ab, 3.39 × 10^12^ AuNP-PEG5000-SH-NH_2_ particles/mL, and 0.125 µM of ZnPcS_4_).

Within both above assays, all control and experimental grouped cell culture plates were then re-incubated for an additional 20 h before being subjected to PDT/PDD laser irradiation. Post-PDT/PDD laser irradiation, the culture plates were re-incubated for another 24 h before being subjected to various biochemical assays.

The PDD assays involved immuno-fluorescent microscope activation using 358Ex/461Em filters to excite any subcellular localized ZnPcS_4_ PS to produce blue fluorescence detectable signals.

Within the PDT assays, cell culture and laser irradiation control plates were exposed to a wavelength of 673 nm (Roithner 1000 mA diode laser, Arryo 4210) in the dark, at a fluency of 10 J/cm^2^ for 16 min and 8 sec.

### 4.4. Chemical Synthesis and Molecular Characterization of the Final Multifunctional Tumor-Targeted Bioactive Nanoconjugate (BNC)

This was done in accordance with the methodology adopted from Naidoo et al., (2019) [8]. Briefly, the ZnPcS_4_ PS was conjugated onto the surface of AuNP-S-PEG5000-NH_2_, which contained 2.85 × 10^15^ AuNPs per mL (Sigma-Aldrich 765309), using spontaneous ligand exchange and adsorption (between Au and PS tetra sulphides) methods. A total of 200 μg/mL of anti-GCC mAb (Abcam: ab122404) was activated using covalent mode carbodiimide crosslinker two-step coupling and bound to the surface of the already conjugated ZnPcS_4_ PS-AuNP-S-PEG5000-NH_2_ using EDC and NHS chemistry in order to form the final BNC (ZnPcS_4_-AuNP-S-PEG5000-NH_2_-Anti-GCC mAb) (Appendix A). The final BNC conjugate was then subjected to various molecular characterization assays, including UV visible and FT-IR spectroscopy, DLS, and ZP, to confirm efficient binding (Appendix A). Additionally, the subcellular localization of the final BNC conjugate was confirmed using immunofluorescent staining.

### 4.5. Subcellular Localization Assays

In order to determine if the final BNC enhanced ZnPcS_4_ PS uptake in CRC CaCo-2 and deterred uptake in WS1 cells, immunofluorescent staining was performed. An intercellular cell adhesion molecule-1 (ICAM-1) is a cellular surface marker membrane protein, which is expressed within almost all in vitro adherent cells, however, different types of cells exhibit different morphological membrane shapes, thereby allowing one to distinguish between them [32]. Thus, ICAM-1 FTIC green immunofluorescent morphological staining was performed to distinguish WS1 fibroblasts (which are spindle shaped) from CRC cells (which are ovoid shaped). DAPI blue immunofluorescent staining was performed to identify blue cellular stained nuclei. The ZnPcS_4_ PS localization within the various control and experimental groups was identified using Cy5 589Ex/610Em microscope fluorescent filters to detect any red auto-fluorescence within cells.

Firstly, CRC CaCo-2 and WS1 cells were seeded at a density of 6 × 10^5^ cells/mL into various control and experimental grouped culture plates, which contained pre-sterilized coverslips. The cultures were treated with a pre-determined dose response of 0.125 μM ZnPcS_4_ PS, ZnPcS_4_-AuNP-S-PEG5000-NH_2_ and BNC in 3 mL culture media. The cells were incubated at 37 °C for 24 h in the dark. After incubation, the cells were fixed with 1 mL of 3.7% (*v*/*v*) formaldehyde solution for 10 min. Thereafter, they were stained for 30 min with 2 µg/mL ICAM-1 mouse monoclonal IgG1 (AbAB2213 AC: Abcam) and 5 µg/mL goat anti-mouse IgG-FITC (AB6785 AC: Abcam) on ice. The cells were then stained with µL of 1 µg/mL DAPI for 5 min at room temperature and rinsed with HBSS. The coverslips were then mounted onto slides and examined using a Carl Zeiss Axio Z1 Observer immuno-fluorescent microscope, with the above discussed various filter fluorescent settings, at 400× magnification. All experimentation was performed in the dark to prevent photobleaching.

### 4.6. Photodiagnostic Assays

To determine if the final actively mAb-targeted BNC could be successfully absorbed by CRC cells only (and not by normal fibroblast cells), as well as be utilized as a fluorescent marker for the early PDD of CRC, immunofluorescent staining was performed.

ICAM-1 FITC green immunofluorescent morphological staining was performed again, in order to distinguish spindle shaped fibroblasts from ovoid shaped CRC cells. However, PDD assays involved immuno-fluorescent microscope activation using 358Ex/461Em filters to excite any subcellular localized ZnPcS_4_ PS to produce blue fluorescence detectable signals.

Initially, CRC CaCo-2 and WS1 cells were seeded at a density of 6 × 10^5^ cells/mL into various experimental and control grouped culture plates, which contained pre-sterilized coverslips. The culture plates were then treated with a pre-determined dose response of 0.125 μM ZnPcS_4_ PS, ZnPcS_4_-AuNP-S-PEG5000-NH_2_ and BNC in 3 mL of culture media. The culture plates were then incubated overnight at 37 °C. Thereafter, cells were fixed with 3.7% paraformaldehyde for 10 min at room temperature. Then the cells were stained with 2 µg/mL ICAM-1 mouse monoclonal IgG1 (AbAB2213 AC: Abcam) and 5 µg/mL goat anti-mouse IgG-FITC (AB6785 AC: Abcam) on ice, at room temperature, for 30 min. After incubation, the cells were rinsed with HBSS. The coverslips were then mounted onto slides and examined using a Carl Zeiss Axio Z1 Observer immuno-fluorescent microscope, with the above discussed various filter fluorescent settings, at 400× magnification. All experimentation was performed in the dark to prevent photobleaching.

### 4.7. Morphological Assessment Post-PDT

Morphological changes in the various CaCo-2 control and experimental groups, post-PDT treatment, were observed and captured using an inverted microscope with a built-in digital camera (Olympus CKX41 C5060-ADUS) at 100× magnification.

### 4.8. Cellular Death Pathway Detection Assay Post-PDT

Post-PDT treatment, the various CaCo-2 control and experimental groups were stained, in accordance with manufacturer instructions, using the Annexin V-FITC/PI cell death detection kit (BD Scientific: BD/556570). This assay was used to determine early or late apoptotic, as well as necrotic, phases of cells death within the various control and experimental groups post-PDT. The Annexin V-FITC/PI-stained cells were analyzed, with control gating, using the BD Accuri™ C6 flow cytometer.

### 4.9. Statistical Analysis

Graphical results representing the mean and standard deviation of the biochemical assays were performed in duplicate within six independent experiments. The Student’s t-test and one-way analysis of variances (ANOVA) was used for normally distributed data, whereas the Mann-Whitney test was used for non-normal distributed data. These tests were used to determine if there were significant differences between the various control and experimental groups, where values in a 95% confidence interval (*p* < 0.05 *, *p* < 0.01 **, or *p* < 0.001 ***) were accepted as statistically significant.

## 5. Conclusions

In conclusion, this study successfully fabricated a bioactive nanoconjugate, ZnPcS_4_-AuNP-S-PEG5000-NH_2_-Anti-GCC mAb (BNC), based upon a PEGylated AuNPs, which was multi-functionalized with a fluorescent PDT metalated ZnPcS_4_ PS, and specific anti-GCC targeting mAbs, to overcome in vitro CRC PDD and PDT challenges. The final BNC was found to be photostable and showed selectively improved subcellular accumulation within targeted CRC in comparison to healthy in vitro cultured cells for successful PDD. Additionally, the final BNC reported significantly higher late apoptotic forms of PDT induced CRC cell death rates (34% ***), when compared to PDT PS administration alone (15% *). The above results suggest that the improved PDD and PDT outcomes were due to the specific PS accumulation within in vitro cultured CRC cells through NP carriage and bioactive anti-GCC targeting. Therefore, this synthesized final BNC platform could possibly improve PDT CRC cancer therapy, due to its targeting abilities, as well as serve as a simultaneous early diagnostic measure in further complementary studies.

In relation to future research, investigations into the cellular signaling pathways, which were affected post-BNC PDT treatment to induce the noted significant forms of apoptotic programmed cell death, require additional understanding. With reference to studies conducted by Turovsky and Varlamova (2021), these types of future studies will contribute to the understanding of the fundamental mechanisms of activation that promote programmed cell death in CRC tissues [55]. Furthermore, similar studies, such as those conducted by Varlamova et al. (2021), require additional research, whereby the sensitivity and intensity of CRC cell death post-PDT, in relation to BNC concentration, is investigated in order to fully elucidate and understand the molecular mechanisms that regulate these PDT cytotoxic effects [56].

## Figures and Tables

**Figure 1 ijms-22-09779-f001:**
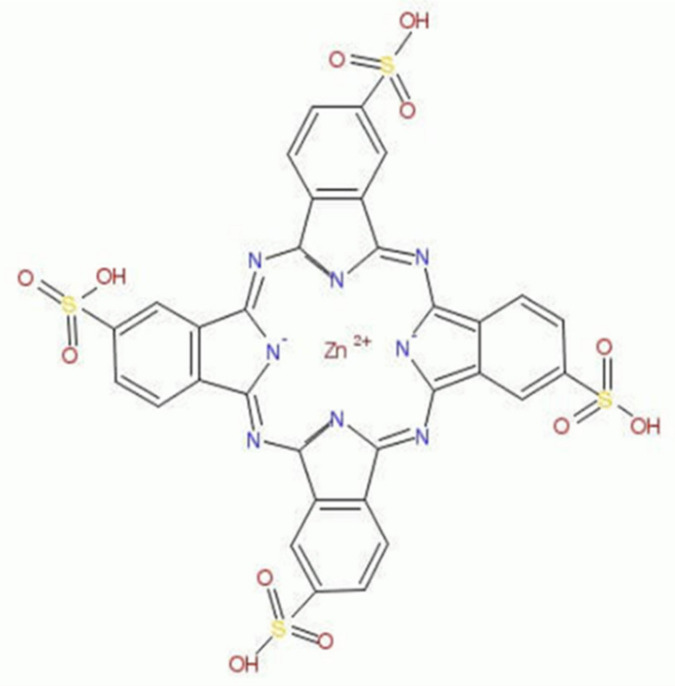
Structure of zinc sulfothiolphthalocyanine (ZnPcS_4_) PS.

**Figure 2 ijms-22-09779-f002:**
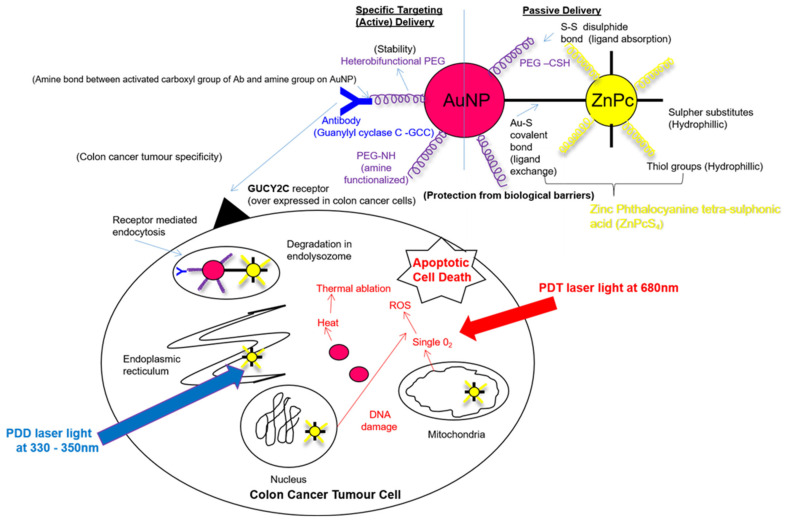
Schematic illustration of the final bioactive nanoconjugate (BNC): ZnPcS_4_-AuNP-S-PEG5000-NH_2_-Anti-GCC mAb assembly.

**Figure 3 ijms-22-09779-f003:**
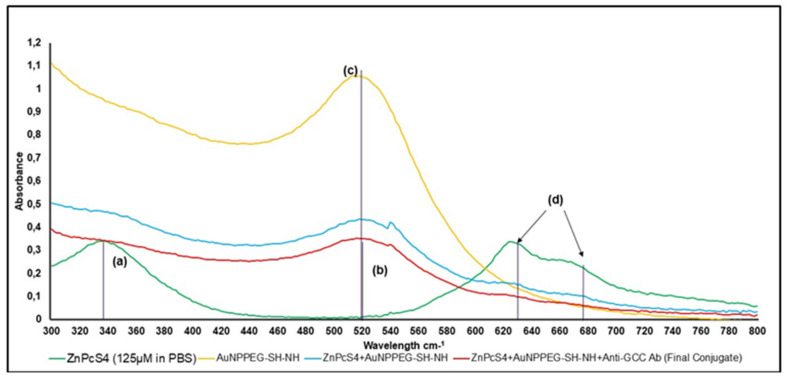
UV visible spectral analysis of BNC and various controls within the 400 to 800 nm visible spectral region. (**a**) PDD ZnPcS_4_ 340 nm 125 µm in PBS, whereas final conjugate ZnPcS_4_ final concentration remained at 125 µm, (**b**) final conjugate 520 nm 0.95 × 10^15^ AuNP particles/mL (2.99-fold fall), (**c**) AuNP-S-PEG5000-NH_2_ 520 nm 2.85 × 10^15^ particles/mL, and (**d**) PDT ZnPcS_4_ 634 and 674 nm 125 µm in PBS, whereas final conjugate is 35 µM ZnPcS_4_ concentration (3.54-fold fall).

**Figure 4 ijms-22-09779-f004:**
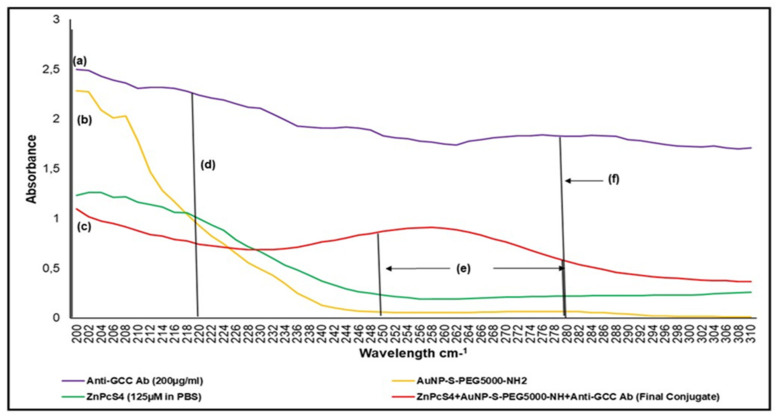
UV-visible protein direct-absorption spectra of the BNC and various controls, acquired between the 200 to 310 nm spectral region, to determine protein presence, as well as amide and disulphide bond confirmation. (**a**) amine NH_2_ 195 nm group on Ab n’ terminal, (**b**) amine NH_2_ 195 nm group on functionalized AuNPPEG-NH_2_, (**c**) amine NH_2_ 195 nm group on final conjugate Ab n’ terminal (Ab functional), (**d**) peptide/amide bond 220 nm of final conjugate (functionalized AuNPPEG-NH_2_ lost amine group to bond with carboxyl c’ terminus of Ab and form a peptide bond), (**e**) disulphide bond 250 to 280 nm (ligand exchange process was performed in the presence of air and, thus, caused ZnPcS_4_ to form a dimer), and (**f**) anti-GCC Ab protein 280 nm 200 µg/mL (6.47 times higher absorbance than final BNC conjugate, thereby suggesting final concentration of Ab within the BNC molecule was 30.91 µg/mL).

**Figure 5 ijms-22-09779-f005:**
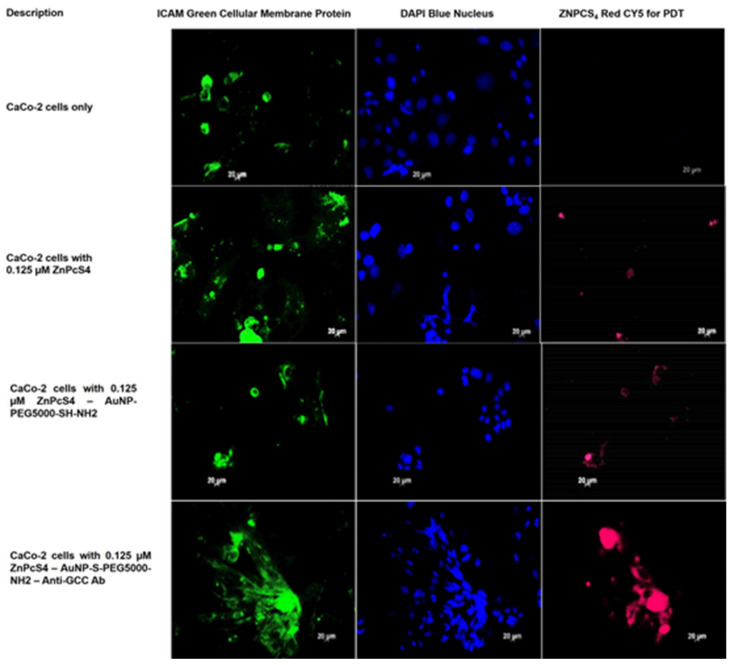
Subcellular localization images comparison of ZnPcS_4_ PS uptake in CaCo-2 cells, treated with ZnPcS_4_ PS alone, AuNP-S-PEG5000-NH_2_-ZnPcS_4_, and the final BNC. DAPI-stained nuclei (blue), cellular ICAM membrane proteins (green), and ZnPcS_4_ PS localization (red).

**Figure 6 ijms-22-09779-f006:**
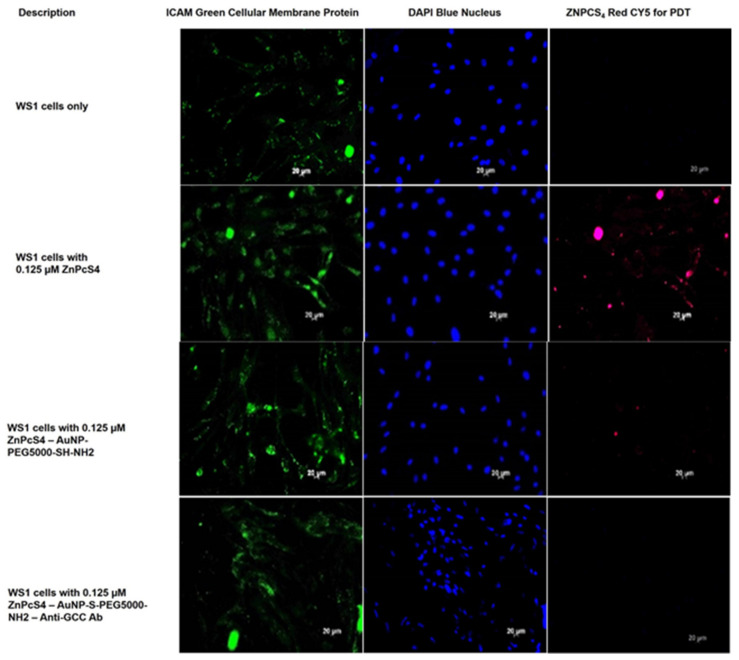
Subcellular localization images comparison of ZnPcS_4_ PS uptake in WS1 cells, treated with ZnPcS_4_ PS alone, AuNP-S-PEG5000-NH_2_-ZnPcS_4_, and the final BNC. DAPI-stained nuclei (blue), cellular ICAM membrane proteins (green), and ZnPcS_4_ PS localization (red).

**Figure 7 ijms-22-09779-f007:**
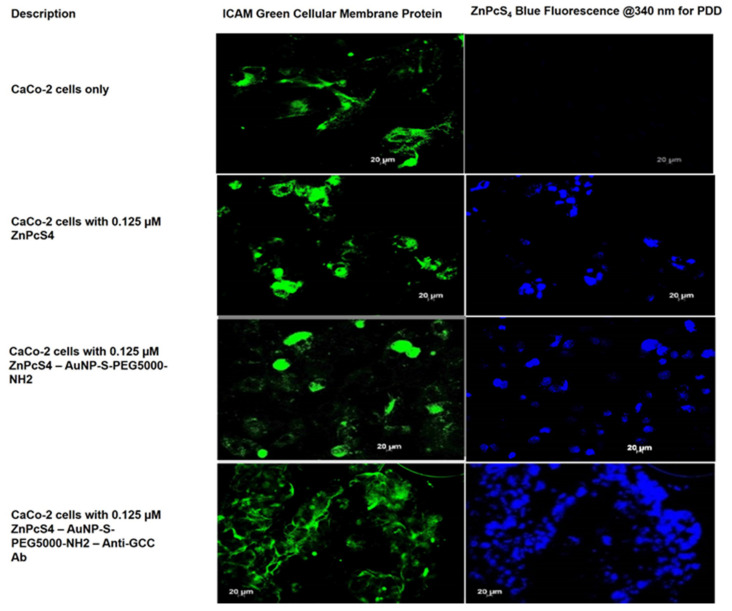
PDD images comparison of ZnPcS_4_ PS uptake in CaCo-2 cells, treated with ZnPcS_4_ PS alone, AuNP-S-PEG5000-NH_2_-ZnPcS_4_, and the final BNC. Cellular ICAM membrane proteins (green), and ZnPcS_4_ PS localization (blue).

**Figure 8 ijms-22-09779-f008:**
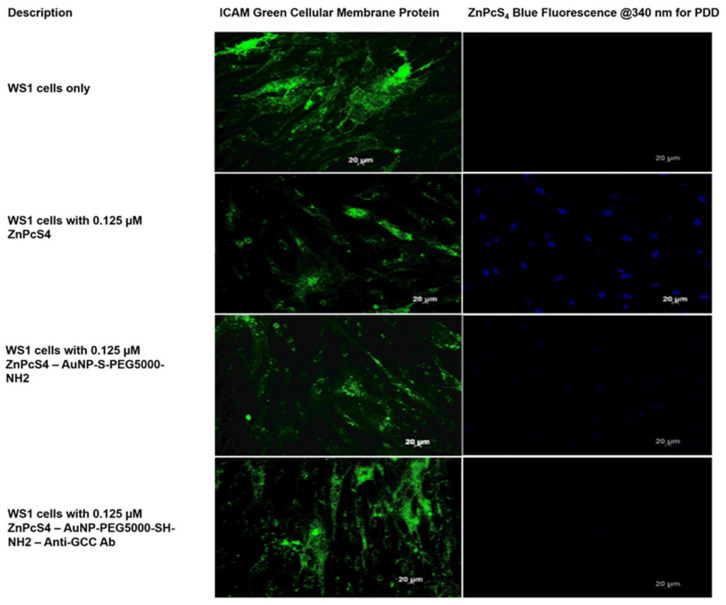
PDD images comparison of ZnPcS_4_ PS uptake in WS1 cells, treated with ZnPcS_4_ PS alone, AuNP-S-PEG5000-NH_2_-ZnPcS_4,_ and the final BNC. Cellular ICAM membrane proteins (green), and ZnPcS_4_ PS localization (blue).

**Figure 9 ijms-22-09779-f009:**
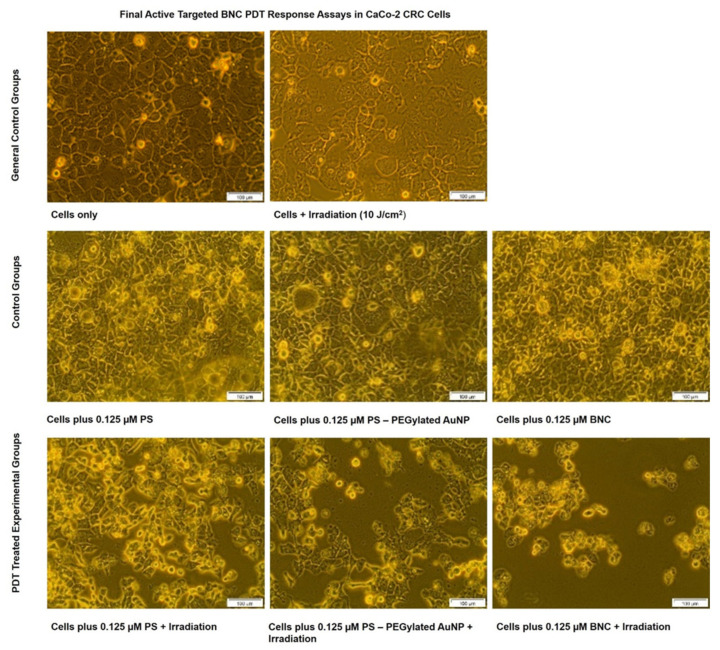
CaCo-2 CRC cells’ morphological comparison of various control and experimental groups within final BNC PDT response assays.

**Figure 10 ijms-22-09779-f010:**
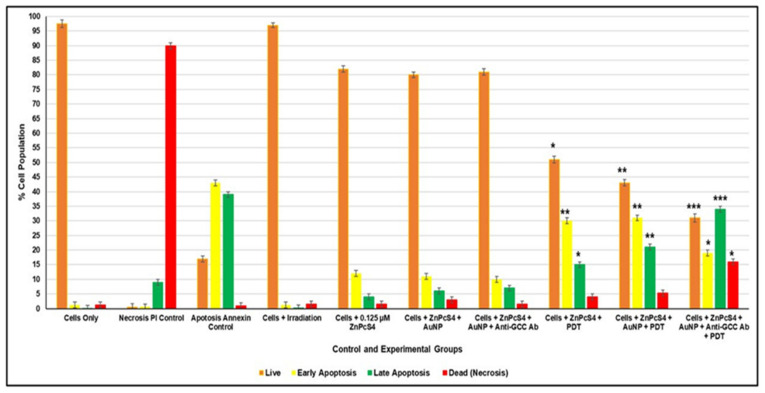
Percentage of different stages of cell death post-PDT using the Annexin V-FITC/PI flow cytometry detection method on various CRC control and experimental groups. The Student’s t-test was used to determine if there were significant differences between the various control and experimental groups, where values in the 95% confidence interval (*p* < 0.05 *, *p* < 0.01 ** or *p* < 0.001 ***) were accepted as statistically significant. The cells-only control was used as the live cell comparable control when statistically evaluating viable cells in experimental groups, the necrosis PI control was used to assess significant cell death within experimental groups, and the apoptosis Annexin control was used to statistically compare early and late forms of apoptotic cell death within the various experimental groupings.

## Data Availability

The data presented in this study is available on request from the corresponding author. The data is not publicly available due to privacy of unpublished data sets.

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
