# Peer review of "Targeted Nanoparticle Photodynamic Diagnosis and Therapy of Colorectal Cancer"

_ijms, 2021, doi:10.3390/ijms22189779_

Round 1
Reviewer 1 Report
The aim of the study is very interesting, topical and original, however, the form of presentation of the results has a number of significant shortcomings.
1.The «Introduction» has been written in too much detail, it needs to be shortened, and some of the information should be moved to the «Discussion» section, which, on the contrary, is briefly described.
- Fig.10 is difficult to understand due to poor quality. In the caption to the figure, it is necessary to indicate data on the statistical processing of the results.
- Translation of the manuscript requires substantial verification, the authors made many grammatical errors. For example, in lines 188, 256, etc. Many sentences are very long, which makes them difficult to understand. For example, in section 2.2.2., line 212.
- How can the authors explain the low absorption capacity of PS by cancer cells Caco-2? Is this related to the morphology and phenotypic features of this cell line?
- Authors should use in the discussion the latest works on the mechanisms of action of nanoparticles: https://www.mdpi.com/1422-0067/22/15/7798; https://doi.org/10.3390/biology10080743
Author Response
Reviewer 1:
Comments and Suggestions for Authors
- The «Introduction» has been written in too much detail, it needs to be shortened, and some of the information should be moved to the «Discussion» section, which, on the contrary, is briefly described. Author: within the previous round of revisions reviewers asked that more information be inserted in the introduction, so authors feel this would now be contradictory to remove. Furthermore reviewer 2 requested within this round of revision requested that further explanation to PDT and PDD be inserted in the manuscripts introduction. Taking into consideration both reviewers comments, no further text was inserted into the introduction and rather the authors have worked on lengthening the discussion and have also inserted a conclusion section to improve upon the manuscripts revision and so satisfy both reviewers requests.
- 10 is difficult to understand due to poor quality. Author: noted and the axis size and font of the Figure 10 has been enhanced, as well as enlarged in the manuscript to improve the quality of the figure.
- In the caption to the figure 10, it is necessary to indicate data on the statistical processing of the results. Author: noted and Figure 10’s caption has been updated to include data on statistical processing of the results.
- “Figure 10. Percentage of different stages of cell death post-PDT using the Annexin V-FITC/PI flow cytometry detection method on various CRC control and experimental groups. The Student t-test was used to determine if there were significant differences between the various control and experimental groups, where values in the 95 % confidence interval (P < 0.05*, P < 0.01** or P < 0.001***) were accepted as statistically significant. The cells only control was used as the live cell comparable control when statistically evaluating viable cells in experimental groups, the necrosis PI control was used to assess significant cell death within experimental groups and the apoptosis annexin control was used to statistically compare early and late forms of apoptotic cell death within the various experimental groupings.”
- Translation of the manuscript requires substantial verification, the authors made many grammatical errors. Author: noted and full English language, style and grammatical changes have been applied throughout the manuscript and supplementary material.
- Many sentences are very long, which makes them difficult to understand:
- For example, in lines 188, 256, etc. Author: noted and shortened sentences.
- For example, in section 2.2.2., line 212. Author: noted and shortened sentence.
- How can the authors explain the low absorption capacity of PS by cancer cells Caco-2? Is this related to the morphology and phenotypic features of this cell line? Author: noted and elaborated within 2.3. Subcellular Localization Assays the explanation for the low PS absorption was probably due to PS aggregation when administered alone.
- Authors should use in the discussion the latest works on the mechanisms of action of nanoparticles: https://www.mdpi.com/1422-0067/22/15/7798; https://doi.org/10.3390/biology10080743. Author: noted and requested references were inserted within the conclusion section of the manuscript, “In relation to future research, investigations into the cellular signalling pathways which were affected post BNC PDT treatment to induce the noted significant forms of apoptotic programmed cell death, require understanding. With reference to studies Turovsky and Varlamova (2021), these types of future studies will contribute to under-standing of the fundamental mechanisms of activation which promote programmed cell death in CRC tissues. Furthermore, similar studies like those conducted by Varlamova et al. (2021), require additional research, whereby the sensitivity and intensity of CRC cell death post PDT, in relation to BNC concentration is investigated, in order to fully elucidate and understand the molecular mechanisms which regulate these PDT cytotoxic effects.”
- Many sentences are very long, which makes them difficult to understand:
Authors note: due to reviewers’ suggestions of requiring additional reference referrals, updated references were added within the manuscript and so all references were renumbered throughout the text.
Reviewer 2 Report
The authors presented the paper "Targeted Nanoparticle Photodynamic Diagnosis and Therapy 2 of Colorectal Cancer"
1) I recommend not to use any abbreviations in the abstract part if it is possible. Also, I think you haven't to use PS - photosensitizer as It looks the same as PcS part, and in some places, you can get confused.
2) Figure 1. I think have to be 4-atom coordinated (with two nitrogens more)
3) Figure 2 looks busy. Maybe It can be divided into two 1) schematic illustration of bioactivity and 2) the advantages, etc. of the construction.
4) more 2-5 years' new publications should be presented in the introduction section to better understand the problem that contributes to the area. I recommend enlarging a bit Photodynamic diagnosis (PDD) and photodynamic therapy (PDT) part presenting the possible molecules used in the area with some of the references.
5) Figure 10 is not readable well. What does * mean - write in Figure 10 description.
6) About SI. Of course, you can present the simple experimental methods in the SI. But I have found there enough information to be presented in the paper such as results.
For example, you write: Trypan Blue Exclusion cell viability assay and Lactate Dehydrogenase (LDH) cytotoxicity and membrane integrity assays results can be found in the supplementary material section 2.1.
The characterization of the samples It is an important thing to obtained good biological results. I think It will be better to place in the SI some more of the experimental part thing (If you want) but to present briefly the SI results and pictures in the paper.
Figures 1 and 2 in the SI are not readable well. Also, I recommend naming the picture there as Figure S1, etc.
7) In the discussion part I haven't found one important thing as comparison of the work conjugates and the literature obtained results in the research area. It is difficult to understand is the obtained results are significant for the area without that data. The same limitation of the work presents in the introduction part as mentioned above.
8) It will be excellent to form the conclusion section which essentially improves the work's main results understanding. Also, It will be nice to present briefly the advantages and limitations in that text part. Moreover, the ideas or suggestions for further investigations will adorn the work.
Author Response
Reviewer 2:
Comments and Suggestions for Authors
- I recommend not to use any abbreviations in the abstract part if it is possible. Also, I think you haven't to use PS - photosensitizer as It looks the same as PcS part, and in some places, you can get confused. Authors: unfortunately, the only way to meet the journal requirements of an abstract length not exceeding 200 words, was to utilize abbreviations. We do note that PS and PcS abbreviations might look similar, however these are standard / conventional photodynamic therapy abbreviations utilized throughout scientific literature and so to change them be inappropriate. Authors wish to leave this decision up to the editor.
- Figure 1. I think have to be 4-atom coordinated (with two nitrogens more). Authors: noted and this image has been corrected.
- Figure 2 looks busy. Maybe It can be divided into two 1) schematic illustration of bioactivity and 2) the advantages, etc. of the construction. Authors: within the previous round of revisions reviewers asked specifically that a combinative figure of this magnitude be inserted into the manuscript. Authors feel that this figure encompasses the assembly of the BNC, as well as effectively represents its purpose and function and so don’t feel the need to break it down into separate illustrations. Authors wish to leave this decision up to the editor.
- more 2-5 years' new publicationsshould be presented in the introduction section to better understand the problem that contributes to the area. I recommend enlarging a bit Photodynamic diagnosis (PDD) and photodynamic therapy (PDT) part presenting the possible molecules used in the area with some of the references. Author: Reviewer 1 within this round of revisions noted that the introduction of this manuscript is too long, so to insert more information would be contradictory. Hence, authors decided to compromise and satisfy this request by inserting updated references and referrals to PDD and PDT within the discussion of this manuscript instead of the introduction.
- Figure 10 is not readable well. Author: noted and the axis size and font of the Figure 10 has been enhanced to improve the quality of the figure.
- What does * mean - write in Figure 10 description. Author: noted and Figure 10’s caption has been updated to include data on statistical processing of the results to explain *, this was however also explained within Section 4.9. Statistical Analysis of the manuscript. “Figure 10. Percentage of different stages of cell death post-PDT using the Annexin V-FITC/PI flow cytometry detection method on various CRC control and experimental groups. The Student t-test was used to determine if there were significant differences between the various control and experimental groups, where values in the 95 % confidence interval (P < 0.05*, P < 0.01** or P < 0.001***) were accepted as statistically significant. The cells only control was used as the live cell comparable control when statistically evaluating viable cells in experimental groups, the necrosis PI control was used to assess significant cell death within experimental groups and the apoptosis annexin control was used to statistically compare early and late forms of apoptotic cell death within the various experimental groupings.”
- About SI. Of course, you can present the simple experimental methods in the SI. But I have found there enough information to be presented in the paper such as results.
- For example, you write: Trypan Blue Exclusion cell viability assay and Lactate Dehydrogenase (LDH) cytotoxicity and membrane integrity assays results can be found in the supplementary material section 2.1. The characterization of the samples It is an important thing to obtained good biological results. I think It will be better to place in the SI some more of the experimental part thing (If you want) but to present briefly the SI results and pictures in the paper. Author: noted however, the authors feel the manuscript already as is contains 10 figures and to insert more would be overwhelming. The trypan blue and LDH assays were merely used as dose response assays to determine the ICD50 for the BNC, and in no way significantly contribute to the final outcomes of the PDD and PDT BNC assays and so should remain in the SI. Additionally, the most important characterization results such as UV etc were inserted in the main manuscript and the confirmatory characterization results such as FTIR, DLS and ZP have been discussed in the main manuscript and should the reader wish to consult a figure they have been referred to in the supplementary section. Authors wish to leave the SI figures where they current are, however if the editor feels they should be inserted into the main manuscript the authors shall do so.
- Figures 1 and 2 in the SI are not readable well. Author: noted and figures have been enlarged, their axis text has been made bigger and bolded for clarity
- Also, I recommend naming the picture there as Figure S1, etc. Author: noted and all figures and tables within the SI captions and text have been referred to as Figure / Table SX.
- In the discussion part I haven't found one important thing as comparison of the work conjugates and the literature obtained results in the research area. It is difficult to understand is the obtained results are significant for the area without that data. The same limitation of the work presents in the introduction part as mentioned above. Author: noted and discussion has been updated to include various comparisons of literature and references obtained with results in this area to highlight the significance of this work.
- It will be excellent to form the conclusion section which essentially improves the work's main results understanding. Also, It will be nice to present briefly the advantages and limitations in that text part. Moreover, the ideas or suggestions for further investigations will adorn the work. Author: noted and a conclusion section which essentially improves the work's main results understanding, with advantages and limitations, as well as suggestions for further investigations has been inserted.
- English language and style are fine/minor spell check required. Author: noted and full English language, style, sentence shortening, and grammatical changes have been throughout the manuscript and supplementary material.
Authors note: due to reviewers’ suggestions of requiring additional reference referrals, updated references were added within the manuscript and so all references were renumbered throughout the text.
Round 2
Reviewer 1 Report
My comments have been eliminated. The article has been significantly improved. I wish the authors the best in further research.
Reviewer 2 Report
Thank you for the revised paper. The paper has been significanly improved.
This manuscript is a resubmission of an earlier submission. The following is a list of the peer review reports and author responses from that submission.
Round 1
Reviewer 1 Report
The manuscript of Simelane et al., titled 'Targeted Nanoparticle Photodynamic Diagnosis and Therapy 2 of Colorectal Cancer,' describes the preparation of a bioactive nanoconjugate: ZnPcS4 - AuNP-PEG5000-SH-NH2 – Anti-GCC mAb (BNC) to analyze the PDD and PDT performance towards CRC.
The work is well organized and clear. I recommend that the contribution can be published after major corrections and questions that raised, followed by suggestions to the text that will be found in the PDF file attached:
- It would be helpful for general readers to have a figure similar to Figure 1 in the reference [8] to visualize the assembly of your NP, plus a clear structure of the ZnPcS4 must be added in the introduction.
- The gold nanoparticle from sigma Aldrich has attached the Thiols to the gold. The nomenclature must be adjusted accordingly to AuNP-S-PEG5000-NH2.
- Would you please revise Figures 1 and 2 (check pdf)
Authors claimed that there is a disulfide bond formation between the PEG-Thiol and ZnPcS4 during the ligand exchange, this type of reaction during ligand exchange is not common (https://etheses.whiterose.ac.uk/1153/3/Chapter_2.pdf), and the formation of disulfide bonds from a thiol already attached to the gold nanoparticle will require the use of, for example, hydrogen peroxide. The authors based this claim on reference 19. However, on ref. 19, the discussion is focused on the dimerization of ZnPcS11 (Figure 2b, ref 19) that forms a disulfide bond between itself. The formation of the dimer explains the two absorptions band (Q bands) around 620 nm and a disulfide bond formation. The authors probably did the ligand exchange in the presence of air overnight, which could allow the dimerization of the ZnPcS4 to form (ZnPcS4)2. This matches with the authors' observations of a disulfide bond (the covalent bond formation), making the AuNP assembly (-S-PEG5000-NH-Anti-GCC mAb + ZnPcS4 + (ZnPcS4)2). The discussion must be updated accordingly.
- A signal at 280 was used to measure the concentration of the assembly. Please re-calculate, not saturating the detector. Prepare a calibration curve (with different concentrations) where a straight line must be observed, and the concentration should be determined using a calibration curve.
- Fluorescence spectra are missing.
- On the FT-IR data, a spectrum of Anti-GCC ab must be measured and compared. Also, could you show the region between 3000 and 3200 cm-1 to observe the behavior of the –NH2 and –NH- vibrational modes?

Reviewer 2 Report
To develop readers' understanding, the author should provide schematic illustration presenting this work.
Detail of each data should be explained not in the graphs but in captions. The data look too messy to understand, please sophisticate all the data.
Initially, the characterization of the BNC is not enough.
1) The author must quantified the amount of conjugated antibody on the surface of GNPs.
2) Fluorescence intensity from conjugated Pc derivatives should be measured. Fluorescence from Pcs are usually quenched via self-aggregation. These eyesights should be included the manuscript.
3) Capacity to generate reactive oxygen species have should be addressed.
In fluorescent microscopic observation, the author have to provide fluorescent data in cell only control if the cells were not exposed to PSs. To conclude the fluorescence are not from autofluorescence but from delivered PSs. (Fig. 3, 4, 5, and 6). In addition, phase contrast should be provided here to develop readers' understandings. If the author want to address subcellular distribution, more magnified images should be required. In my opinion, it's really tough to conclude distribution of delivered PSs from these data.
In immunofluorescent study, Caco-2 and WS1 cells were used as GCC positive cell and GCC negative cell, respectively. However, there are no evidences and/or references that support authors' claims. To clarify the authors' claim, the author have to quantify the expression level of GCC by RT-PCR and/or consult with several references. It is not enough to conclude your claim from current form.